# LoCo: Learning 3D Location-Consistent Image Features with a Memory-Efficient Ranking Loss

**Dominik A. Kloepfer**
Visual Geometry Group
University of Oxford
dominik@robots.ox.ac.uk

**João Henriques**
Visual Geometry Group
University of Oxford
joao@robots.ox.ac.uk

**Dylan Campbell**
School of Computing
Australian National University
dylan.campbell@anu.edu.au

## Abstract

Image feature extractors are rendered substantially more useful if different views of the same 3D location yield similar features while still being distinct from other locations. A feature extractor that achieves this goal even under significant viewpoint changes must recognise not just semantic categories in a scene, but also understand how different objects relate to each other in three dimensions. Existing work addresses this task by posing it as a patch retrieval problem, training the extracted features to facilitate retrieval of all image patches that project from the same 3D location. However, this approach uses a loss formulation that requires substantial memory and computation resources, limiting its applicability for large-scale training. We present a method for memory-efficient learning of location-consistent features that reformulates and approximates the smooth average precision objective. This novel loss function enables improvements in memory efficiency by three orders of magnitude, mitigating a key bottleneck of previous methods and allowing much larger models to be trained with the same computational resources. We showcase the improved location consistency of our trained feature extractor directly on a multi-view consistency task, as well as the downstream task of scene-stable panoptic segmentation, significantly outperforming previous state-of-the-art.

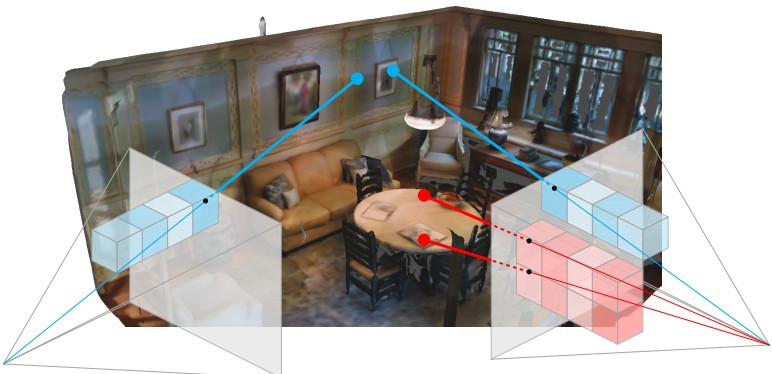

Figure 1: Our approach—LoCo—offers memory-efficient learning of location-consistent (LoCo) features. That is, features that backproject to nearby 3D locations are encouraged to have similar image patch features (illustrated here by the pair of blue stacked cubes and the pair of red stacked cubes), while those that backproject to well-separated points are trained to have more dissimilar features (here, the blue *vs.* red cube stacks). This is achieved via a novel ranking loss that reformulates and corrects the smooth average precision loss proposed in previous work [4, 26]. This facilitates the derivation of a close approximation to the loss that is significantly more efficient to compute, allowing the method to scale to much larger models with the same computational resources.

38th Conference on Neural Information Processing Systems (NeurIPS 2024).

# 1 Introduction

Reasoning in 3D is critical for developing a useful visual understanding of an environment. However, image-centric approaches, including patch feature extractors like DINO [5, 31], are not 3D-consistent, as a recent paper by El Banani *et al.* [12] demonstrates. That is, the same 3D location may yield significantly different features from different viewpoints in space and time, due to occlusions, self-occlusion, reflections, lighting variations, and motion. When this occurs, it is challenging to maintain a spatially and temporally consistent model of the world. Thus, it is useful to first convert visual observations into a form that is stable across different spatio-temporal viewpoints.

Existing approaches [3, 18, 27, 29, 40] aggregate or distil visual features in 3D, ensuring view-consistency at the cost of requiring a full 3D reconstruction pipeline. In contrast, recent work [26] explores a more flexible image-centered representation that encourages similarity between image patch features that backproject to the same region of 3D space, within a spatial tolerance. It employs a ranking-based loss function, smooth Average Precision (sAP), to encourage all spatially co-located features to be similar and all non-colocated features to be dissimilar, as illustrated in Fig. 1.

Like this approach, we formulate the learning problem as one of patch retrieval: given one image patch, retrieve with high precision and recall all patches in other views that project from the same 3D region. However, the smooth Average Precision (sAP) loss function requires substantial memory and computation, precluding its deployment in large-scale training. By rewriting the loss function in terms of pairs of image patches rather than individual patches, we derive a more general form of the sAP objective that lends itself to approximation. This novel formulation enables improvements in memory efficiency by three orders of magnitude, mitigating a key bottleneck and allowing larger models to be trained with the same computational resources. By applying this novel loss function within a new training strategy, we obtain a method for memory-efficient learning of location-consistent (LoCo) features that are semantically-meaningful and stable across viewpoints. Our contributions are:

1. A novel reformulation and approximation of the smooth average precision loss function that can be computed significantly more efficiently than the original;
2. A training strategy for scalable and memory-efficient learning of location-consistent image features; and
3. Applications to pixel correspondence estimation and scene-stable panoptic segmentation.

The approach is evaluated on a recently proposed multi-view consistency task [12] and is tested on two real-world indoor datasets, significantly outperforming state-of-the-art feature extractors.

# 2 Related Work

The topics of panoptic image segmentation [25], visual place recognition [20, 23], image retrieval [2, 4, 13, 32], and visual feature learning [5, 16] are well-studied. Here, we focus on the most recent and related work.

**Image segmentation.** Visual features from pre-trained models were shown to be very useful for panoptic (instance and semantic) image segmentation [5, 25, 43]. Interestingly, location-consistent visual features unlock the possibility of "scene-stable panoptic segmentation", where instance IDs are consistent across multiple views of the same scene [26]. This is related to identity-preserving video segmentation [24], where object identities are tracked, leveraging temporal smoothness.

**Image retrieval.** Learning representations that facilitate the ranking of images according to their relevance to a query has been studied extensively [2, 13, 32]. One class of approaches, metric learning, uses contrastive losses [8, 42] to encourage positive instances to be close and negative instances to be further apart, while others optimise ranking-based metrics like Average Precision (AP) directly [4, 34]. For example, Smooth-AP [4] recommends the use of an approximated AP ranking function, which targets the correct ranking without being concerned with the absolute feature distances. We pose our location-consistent feature learning problem as a patch retrieval problem, allowing us to adapt strategies from the image retrieval literature.

**Self-supervised visual feature learning.** Self-supervision has emerged as a dominant strategy for training foundation models in computer vision on large-scale image datasets. Notably, DINO [5] leveraged a knowledge-distillation framework to learn to extract semantically meaningful feature maps. Building upon DINO's foundation, DINOv2 [31] refines the training algorithm and scales up model sizes on larger datasets, resulting in enhanced performance. Another noteworthy approach

is the Masked Autoencoder (MAE) [17], which employs an autoencoder architecture to reconstruct masked-out patches, and demonstrated the scalability of autoencoders to large datasets.

In a similar vein, CroCo [43] and its successor, CroCo-v2 [44], also adopt an encoder–decoder architecture for reconstructing masked-out patches. However, in these works the decoder reconstructs the patches based on a feature map extracted from a non-masked image of the same scene but observed from a different viewpoint. While we also use different viewpoints as the source of the supervision signal like CroCo, the learning process is quite different and makes more explicit use of geometric constraints in forming positive and negative pairs of patches (*cf*. Section 3.2). Most closely related to our work is LoCUS [26], which uses a similar problem set-up and loss function. Where they approach the task from the perspective of extracting distinctive landmarks (individual patches or points in the scene), our areas of interest are *pairs* of patches. Crucially, this allows us to massively decrease the algorithm's memory consumption and unlock significant performance gains. We provide further detail in Section 3.

Other works [27, 29, 40] distil visual features in 3D in order to ensure location-consistency, the same goal as our approach. For example, N3F [40] distilled DINO image features into a 3D feature field using the same rendering loss as NeRF [30]. While the resulting features are 3D-consistent by design, this comes at the cost of requiring a full 3D reconstruction pipeline. In contrast, our approach is image-centric and lightweight, predicting location-consistent features from one image at a time without requiring input poses or re-training for new scenes.

## 3 Efficiently Learning LoCo Image Features

In this section, we outline our method for learning 3D location-consistent (LoCo) image features in a scalable, memory-efficient way. We first formalise the problem definition and define the positive and negative sets, then we reformulate the retrieval objective function and show how this facilitates very significant reductions in the memory requirements. This allows our method to overcome a critical bottleneck in the learning process for these features, permitting us to scale up the model size. Sec. 4 shows that this has a large impact on performance, justifying the need for the reformulation.

### 3.1 Problem Definition

Our goal is to extract feature maps so that the extracted features are similar for sets of image patches that depict the same region in 3D space. Like Kloepfer *et al.* [26], we make this task tractable by recognising that feature maps solving this task can also be used for *patch retrieval*: given an image patch and associated feature vector, retrieve all patches that project from the same region in 3D space. This allows us to adapt methods from the extensive literature on image retrieval.

Formally, given a set $\mathcal{I}$ of training images $I_i \in \mathbb{R}^{H \times W \times 3}$ drawn from an environment $e$, we aim to train a feature extractor $\phi : \mathcal{I} \to \mathbb{R}^{h \times w \times d}$. Here, $h \leq H$ and $w \leq W$ are the height and width of the extracted feature map, and $d$ is the dimension of the extracted feature vectors, so each feature vector is associated with an image patch $x_k \subseteq I_i$ consisting of $\frac{H}{h} \times \frac{W}{w}$ pixels.

We denote an image patch as $x_k$, the associated feature vector as $\theta_k$, a patch pair as $c_\alpha = (x_i, x_j)$, and the associated cosine similarity score between the feature vectors of a patch pair as $s_\alpha = \theta_i^\top \theta_j / (\|\theta_i\| \|\theta_j\|)$. We use Greek subscripts to index *pairs of patches*, rather than individual patches.

Each patch $x_k \in I_i$ is also associated with a point $p_k \in \mathbb{R}^3$ in the environment, obtained by backprojecting the centre coordinates of the patch into the environment using estimated or provided camera intrinsic and extrinsic parameters and depth [15]. This auxiliary information is only required at training time; at test time the trained feature extractor $\phi$ only requires a raw RGB image.

### 3.2 Positive and Negative Sets

As illustrated in Fig. 2a, we define two patches $x_i$ and $x_j$, not necessarily from the same image, as a 'positive' pair if and only if they are drawn from images in the same environment and the distance between their associated 3D points $p_i$ and $p_j$ is below a threshold $\rho$. Therefore, given the index of the environment $e_i$ associated to each patch $x_i$, the set of all positive pairs is

$$\mathbb{P} = \{(x_i, x_j) : \|p_i - p_j\| \leqslant \rho \wedge e_i = e_j\}. \tag{1}$$

Pairs of patches that do not observe the same 3D region form a 'negative' pair. As a form of hard negative mining, we restrict our attention to those negative pairs whose patches depict locations

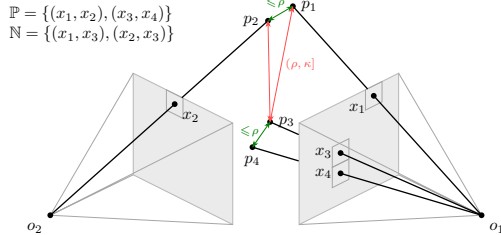

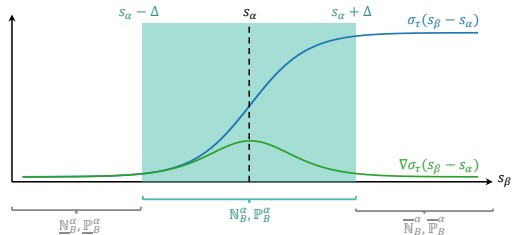

(a) Defining the positive and negative sets.  (b) Memory-efficient strategy for smooth AP.

Figure 2: **(a)** The distance between 3D points $p_i$ associated with patch pairs $(x_i, x_j)$ in the positive set is less than $\rho$, denoted by the green arrows connecting them. The distance between those in the negative set is in $(\rho, \kappa]$, denoted by the red arrows connecting them. All unconnected 3D point pairs are in neither the positive nor negative set, since they are separated by a distance greater than $\kappa$. **(b)** When the absolute similarity difference $|s_\beta - s_\alpha|$ between image pair $\alpha$ and pair $\beta$ is large, the sigmoid in the loss function (blue curve) becomes saturated and does not impact learning. We can avoid the memory cost of back-propagation in these cases by separating the positive $\mathbb{P}$ and negative $\mathbb{N}$ pairs into 3 subsets: saturated below, unsaturated, and saturated above. We choose the saturation threshold $\Delta$ such that the sigmoid gradient there is 0.2% of its maximum value.

within a distance of $\kappa > \rho$ of each other, and define the set of negative pairs as $\mathbb{N} = \{(x_i, x_j) : \rho < \|p_i - p_j\| \leqslant \kappa \wedge e_i = e_j\}$. For convenience we also define the set of all training pairs $\Omega = \mathbb{P} \cup \mathbb{N}$.

For a given patch $x_i$, the thresholds $\rho$ and $\kappa$ define a positive and a negative region around the associated 3D point $p_i$. All other patches that observe points inside the radius-$\rho$ sphere centred at $p_i$ will form a positive patch pair with $x_i$. Likewise, all other patches that observe points outside this sphere but inside the radius-$\kappa$ sphere centred at $p_i$ will form a negative patch pair with $x_i$. The features of the positive pairs are encouraged to be more similar than those of the negative pairs.

It should also be noted that the finite size of these regions means that the training is robust to noise in the depth maps and camera poses used to compute the patch locations in 3D. Noisy patch locations effectively only slightly change the size of these regions and so we expect them not to fundamentally alter the learning algorithm. This could open the door to using less accurate estimates for depth and camera poses in future, which may be easier to obtain, in particular for large-scale datasets.

### 3.3 A Ranking Loss Function for Patch Retrieval

The smooth average precision (AP) loss function, originally introduced in Brown *et al.* [4], was adapted to the setting of patch retrieval around "tentative 3D landmarks" by Kloepfer *et al.* [26], resulting in a vectorised form of the loss function. We streamline this setting by eliminating the need for these landmarks, and instead focus exclusively on retrieving positive pairs of patches.

The aforementioned vectorised smooth AP loss function can be rewritten in terms of patch pairs as

$$
\begin{aligned}
\mathcal{L}(\Omega; \tau, \rho, \kappa) &= -\frac{1}{|\mathbb{P}|} \sum_{c_\alpha \in \mathbb{P}} \frac{1 + \sum_{c_\beta \in \mathbb{P} \backslash \{c_\alpha\}} \sigma_\tau(s_\beta - s_\alpha)}{1 + \sum_{c_\gamma \in \Omega \backslash \{c_\alpha\}} \sigma_\tau(s_\gamma - s_\alpha)} \\
&= -\frac{1}{|\mathbb{P}|} \sum_{c_\alpha \in \mathbb{P}} \frac{1 + \sum_{c_\beta \in \mathbb{P} \backslash \{c_\alpha\}} \sigma_\tau(s_\beta - s_\alpha)}{1 + \sum_{c_\beta \in \mathbb{P} \backslash \{c_\alpha\}} \sigma_\tau(s_\beta - s_\alpha) + \sum_{c_\gamma \in \mathbb{N}} \sigma_\tau(s_\gamma - s_\alpha)},
\end{aligned}
\tag{2}
$$

where $\sigma_\tau(x) = (1 + \exp(-x/\tau))^{-1}$ is the sigmoid function with temperature $\tau$.

This loss function computes a differentiable approximation to the average precision of a binary classifier that classifies pairs of patches as positive or negative based on the similarity of each pair. This approximation becomes exact as $\tau \to 0$ and the sigmoid approaches the indicator function. For each positive pair $c_\alpha$, Eq. (2) calculates the ratio of the rank of $c_\alpha$ among all positive pairs and its rank among all pairs (positive and negative) when ranking the pairs by decreasing similarity. More details can be found in Brown *et al.* [4].

Compared with standard contrastive losses, like a triplet loss [19] or SimCLR [7], the gradient $\partial \mathcal{L}/\partial s_\alpha$ of this ranking loss with respect to the similarity of a positive pair $c_\alpha$ will disappear as

soon as $s_\alpha$ is higher than the similarities of all negative pairs. That is, it does not force $s_\alpha \to +1$ for positive pairs and $s_\alpha \to -1$ for negative pairs, it merely encourages some boundary to exist somewhere between positive and negative pair similarities. This "gentler" contrastive characteristic greatly aids training convergence [26].

## 3.4 Correction Terms for the Batched Loss Function

The sums in Eq. (2) run over all pairs of (positive) patches in the training set. Clearly, computing the exact loss function is completely infeasible for large datasets. However, when sampling batches of positive and negative pairs, correction terms are needed to make the expectation value of the batched loss equal to the exact (unbatched) loss. In particular, some care needs to be taken since each term in the loss function depends on multiple different samples within the batch. Using the subscript $B$ to refer to the batched versions of the sets $\mathbb{P}_B \subset \mathbb{P}$ and $\mathbb{N}_B \subset \mathbb{N}$ of positive and negative pairs, the batched version of our loss function becomes

$$\mathcal{L}_B = -\frac{1}{|\mathbb{P}_B|} \sum_{c_\alpha \in \mathbb{P}_B} \frac{1 + \frac{|\mathbb{P}|}{|\mathbb{P}_B|} \sum_{c_\beta \in \mathbb{P}_B \setminus \{c_\alpha\}} \sigma_\tau(s_\beta - s_\alpha)}{1 + \frac{|\mathbb{P}|}{|\mathbb{P}_B|} \sum_{c_\beta \in \mathbb{P}_B \setminus \{c_\alpha\}} \sigma_\tau(s_\beta - s_\alpha) + \frac{|\mathbb{N}|}{|\mathbb{N}_B|} \sum_{c_\gamma \in \mathbb{N}_B} \sigma_\tau(s_\gamma - s_\alpha)}. \quad (3)$$

The correction factors $\frac{|\mathbb{P}|}{|\mathbb{P}_B|}$ and $\frac{|\mathbb{N}|}{|\mathbb{N}_B|}$ are necessary to ensure that the expectation value of the batched loss is as close as possible to the loss computed across the entire dataset. This is automatically the case for standard loss functions that average over a per-sample loss, due to the linearity of the expected value. However, since the ranking loss computes the (non-linear) ratio of expectations over samples, this linearity is lost. Each loss term depends on multiple pairs, and the $+1$ terms in numerator and denominator introduce additional complications for finding an unbiased estimator. For a detailed derivation of Eq. (3) we refer the reader to Appendix B, where we show that it is a ratio estimator [37] that is simple to calculate and consistent, but has a bias of order $O(1/|\Omega_B|)$.

We note here that neither Brown *et al.* [4] nor Kloepfer *et al.* [26] include these correction factors, which cause the losses in those works to deviate even further from the desired average precision approximation. We quantify this deviation in the appendix.

## 3.5 Improving the Memory Efficiency

A key bottleneck of the loss in Eq. (3) is the memory consumption of the $|\mathbb{P}_B| \times (|\mathbb{P}_B| + |\mathbb{N}_B|)$ matrix containing all the values of $s_\beta - s_\alpha$ and $s_\gamma - s_\alpha$, and the associated computation graph. Since each occurrence of the similarity of a particular patch pair in the loss function only provides a supervision signal for the feature vectors of two individual patches, the batches of positive and negative patch pairs need to be quite large (Kloepfer *et al.* [26] use $|\mathbb{P}_B| \approx 13,000$ and $|\mathbb{N}_B| \approx 100,000$).

To alleviate this problem, we design two ways to significantly reduce the memory consumption of this matrix. First, we observe that the positive pairs $c_\alpha$ in Eq. (3) do not need to be drawn from the same subset of all positive pairs as the ones in $c_\beta$. As long as both are sampled uniformly from the set of all positive pairs, the expected value of the batched loss continues to equal the non-batched loss of Eq. (2). Sampling $c_\alpha$ from a small set of positive pairs $\mathbb{P}'_B \subset \mathbb{P}$ and $c_\beta$ from a large set $\mathbb{P}_B \subset \mathbb{P}$, with $|\mathbb{P}'_B| \ll |\mathbb{P}_B|$, reduces the size of the matrix of similarity differences to $|\mathbb{P}'_B| \times (|\mathbb{P}_B| + |\mathbb{N}_B|)$: only the second dimension is large, while previously both dimensions were large and of comparable size. At the same time, this still retains a supervision signal for a large number of feature vectors, namely those used to construct the larger sets $\mathbb{P}_B$ and $\mathbb{N}_B$.

Second, we observe that, in practice, a large number of the computed similarity differences saturate the sigmoid function. That is, when $|s_\beta - s_\alpha| \gg 0$, the gradient of the sigmoid $\nabla \sigma_\tau(s_\beta - s_\alpha)$ vanishes: these terms make no material difference to the loss gradient. We can use this fact to significantly reduce the number of similarity differences in the computation graph. To do so, we set a threshold $\Delta > 0$ and use the approximation, visualised in Fig. 2b,

$$\sigma_\tau(s_\mu - s_\nu) \approx \begin{cases} 1 & \text{if } s_\mu - s_\nu > \Delta \\ \sigma_\tau(s_\mu - s_\nu) & \text{if } |s_\mu - s_\nu| \leq \Delta \\ 0 & \text{if } s_\mu - s_\nu < -\Delta \end{cases} \quad (4)$$

to divide the uniformly sampled sets of patch pairs $\mathbb{P}_B$ and $\mathbb{N}_B$ into three subsets for each $c_\alpha \in \mathbb{P}'_B$,

$$\mathbb{N}_B = \underline{\mathbb{N}}_B^\alpha \cup \mathbb{N}_B^\alpha \cup \overline{\mathbb{N}}_B^\alpha \tag{5}$$

$$\overline{\mathbb{N}}_B^\alpha = \{c_\beta \in \mathbb{N}_B : s_\beta - s_\alpha > \Delta\}, \tag{6}$$

$$\mathbb{N}_B^\alpha = \{c_\beta \in \mathbb{N}_B : |s_\beta - s_\alpha| \leq \Delta\}, \tag{7}$$

$$\underline{\mathbb{N}}_B^\alpha = \{c_\beta \in \mathbb{N}_B : s_\beta - s_\alpha < -\Delta\}, \tag{8}$$

and similarly for $\mathbb{P}_B$. We can now significantly reduce the number of similarity differences in the computation graph by only including patch pairs from $\mathbb{N}_B^\alpha$ and $\mathbb{P}_B^\alpha$ in the loss computation, to $\sum_{c_\alpha \in \mathbb{P}'_B}(|\mathbb{P}_B^\alpha| + |\mathbb{N}_B^\alpha|)$ in total. Computing these subsets for each $c_\alpha$ still requires us to calculate all $|\mathbb{P}'_B| \times (|\mathbb{P}_B| + |\mathbb{N}_B|)$ similarity differences, but because most pairs' gradients are close to 0 they are not used to compute the loss or any gradients, so these parts of the computational graph can be deleted, leading to substantial memory savings.

To compensate for the non-uniform sampling of $\mathbb{P}_B^\alpha$ and $\mathbb{N}_B^\alpha$, we also need to add additional correction terms to our loss function. Our final memory-efficient, batched loss function is given by

$$\mathcal{L}_B = -\frac{1}{|\mathbb{P}'_B|} \sum_{c_\alpha \in \mathbb{P}'_B} \frac{1 + \frac{|\mathbb{P}|}{|\mathbb{P}_B|} \sum_{c_\beta \in \mathbb{P}_B^\alpha} \sigma_\tau(s_\beta - s_\alpha) + \delta_\alpha^+}{1 + \frac{|\mathbb{P}|}{|\mathbb{P}_B|} \sum_{c_\beta \in \mathbb{P}_B^\alpha} \sigma_\tau(s_\beta - s_\alpha) + \frac{|\mathbb{N}|}{|\mathbb{N}_B|} \sum_{c_\gamma \in \mathbb{N}_B^\alpha} \sigma_\tau(s_\gamma - s_\alpha) + \delta_\alpha^+ + \delta_\alpha^-} \tag{9}$$

with the correction terms given by

$$\delta_\alpha^+ = |\overline{\mathbb{P}}_B^\alpha| \frac{|\mathbb{P}|}{|\mathbb{P}_B|}, \text{ and } \delta_\alpha^- = |\overline{\mathbb{N}}_B^\alpha| \frac{|\mathbb{N}|}{|\mathbb{N}_B|}. \tag{10}$$

For a detailed derivation of these correction terms, as well as a derivation of an upper bound on the error due to the approximation in Eq. (4), we refer the reader to Appendix C.

Restricting our sampling of patch pairs to those whose similarities fall within a certain range is reminiscent of hard negative mining, as previously employed in the context of contrastive learning by, *e.g.*, Robinson *et al*. [33]. However, standard approaches are not easily applicable to our loss function since it operates on *pairs* rather than individual samples. Furthermore, both motivation and effect of our approach differs from hard negative mining methods. We discuss this further in Appendix E, where we also compare empirically with standard contrastive learning and hard negative mining.

## 4 Experiments

In this section, we present our experiments, where we evaluate the performance of LoCo features at a multi-view consistency task and at scene-stable panoptic segmentation.

### 4.1 Experimental Setup

The training dataset we use comprises 59 environments of the Matterport3D dataset, resizing the images to $256 \times 320$ pixels. The Matterport3D dataset is particularly suitable for our task of enforcing multi-view consistency due to its diversity and the way it captures varied viewpoints of the same scene through panorama cropping. Datasets such as ScanNet [10] provide less viewpoint variation per scene due to their trajectory-based data collection. Contrary to Kloepfer *et al*. [26], we use all available images in the Matterport3D training scenes instead of restricting to images taken in the horizontal plane.

Due to limited computational resources, we were unable to use our loss function to train a full foundation model from scratch. Instead, we adapt the architecture used by DINO-Tracker [41], keeping pre-trained DINO [5] features frozen and training a convolutional neural network to learn additive residuals to those features. We use values of $\rho = 0.5$m for the positive radius, $\kappa = 5.0$m for the negative radius, $\tau = 0.01$ for the sigmoid temperature, and $\Delta = 0.076$ for the saturation threshold. With these values, $\sigma_\tau(\Delta) = 0.9995$ and the gradient is $0.2\%$ of the maximum gradient of the sigmoid function, making this a conservative choice with little impact on the training dynamics. The threshold $\Delta$ is a hyperparameter that can be decreased to obtain further memory savings, at the expense of some performance decrease. We provide further implementation details regarding the efficient sampling of patch pairs in Appendix D, and will publicly release our training code.

Table 1: Results on the pixel correspondence task on the Paired split [36] of ScanNet [10], as introduced by El Banani *et al.* [12]. We report the recall of accurate pixel correspondences at a reprojection error threshold of 10 pixels, for image pairs with the respective viewpoint changes. We also report the GPU Memory required for training LoCUS-based and LoCo models (for LoCUS we use the authors' values). $^\dagger$ uses 64-dimensional feature vectors.

| Model | GPU Memory | $0°–15°$ | $15°–30°$ | $30°–60°$ | $60°–180°$ |
|---|---|---|---|---|---|
| LoCUS$^\dagger$ [26] | 11GB | 23.5 | 18.8 | 13.5 | 7.5 |
| DINO [5] | | 45.0 | 34.3 | 22.6 | 10.7 |
| DINOv2 [31] | | 37.0 | 27.5 | 19.7 | **11.2** |
| DINOv2 [31], Blocks 1–6 | | 47.1 | 36.4 | 22.4 | 8.4 |
| CroCo-v2 [44] | | 16.8 | 12.4 | 7.4 | 3.7 |
| LoCo ($\Delta = 0.076$) (Ours) | 48GB | **61.8** | **52.7** | **31.8** | 10.3 |
| LoCo ($\Delta = 0.053$) | 42GB | 59.9 | 49.8 | 29.1 | 9.5 |
| LoCo ($\Delta = 0.029$) | 40GB | 57.9 | 47.5 | 28.1 | 9.6 |
| LoCo (w/ DINOv2 backbone) | 38GB | 58.2 | 49.2 | 30.0 | 10.5 |
| LoCo (w/ LoCUS architecture)$^\dagger$ | 6GB | 27.8 | 18.0 | 12.1 | 6.9 |

## 4.2 Baselines

We compare our LoCo features to those from several state-of-the-art feature extractors, as well as models specialised to the downstream segmentation task.

**DINO [5].** We use the pre-trained ViT-base model and extract 768-dimensional features for each patch of $8 \times 8$ pixels by discarding the class token and reshaping the output of the final Transformer-Block into a feature map.

**DINOv2 [31].** We use the pre-trained ViT-base model and extract 768-dimensional features for each patch of $14 \times 14$ pixels by again discarding the class token and reshaping the output of the final Transformer-Block. We generally use higher-resolution images as inputs for this model to extract feature maps of the same shape as for LoCo and DINO features.

**CroCo-v2 [44].** We use the ViT-base encoder that was pre-trained with the Base-Decoder and extract 768-dimensional features for each patch of $16 \times 16$ pixels, again using higher-resolution as inputs.

**LoCUS [26].** We use the pre-trained LoCUS weights that are publicly available for a landmark-radius of 0.2m. We use it to extract 64-dimensional feature vectors for each patch of $8 \times 8$ pixels.

**MaskDINO [28].** For the scene-stable object segmentation task (*cf.* Section 4.4) we also compare with MaskDINO, a state-of-the-art specialised panoptic segmentation algorithm. It however is not designed to recognise the same object in different images. We therefore match the per-image object indices produced by the algorithm to the ground-truth per-scene object index whose mask has the highest IoU with the object mask in question.

## 4.3 Multi-View Consistency

We first directly evaluate the location consistency of the features extracted by different models across different views. To do so, we follow the protocol introduced by El Banani *et al.* [12] to test the multi-view consistency of feature extractors on a pixel correspondence estimation task. Briefly, given a pair of images, we extract a fixed number of pixel matches by filtering the nearest neighbour matches using a ratio test. For more details, the reader is referred to the original paper [12].

Like El Banani *et al.*, we evaluate on the Paired ScanNet [10] split proposed by Sarlin *et al.* [36], reporting the recall at a reprojection error threshold of 10 pixels for different viewpoint changes in Table 1. As we can see, our model outperforms the baselines by a significant margin. We also see that DINOv2 [31] performs worse than the original DINO [5]. This situation is somewhat reversed when sourcing the features from the first six Transformer blocks (instead of the final feature map). This suggests that while earlier layers of DINOv2 are still reasonably location-consistent, later layers create more semantically meaningful features that accordingly do not vary much by the patches' location in 3D space, explaining why they perform worse in this setting. CroCo-v2 [44] performs relatively poorly on this task, despite its training objective being explicitly designed for multi-view

Table 2: Scene-stable panoptic segmentation results on unseen Matterport3D [6] and ScanNet [10] environments. Except for MaskDINO, each method extracts $d$-dimensional feature vectors for $30 \times 40$ patches that are then classified into a scene-wide object index using a linear probe. The feature dimension is $d = 768$, except for LoCUS ($d = 64$) due to its high memory consumption. *Per-image instance indices are matched to the ground-truth per-scene indices based on mask IoU.

| Model | $d$ | Matterport3D | | | ScanNet | | |
|---|---|---|---|---|---|---|---|
| | | Jac | IoU | AP | Jac | IoU | AP |
| LoCUS [26] | 64 | 28.6 | 29.6 | 40.5 | 68.9 | 59.2 | 68.7 |
| DINO [5] | 768 | 65.2 | 65.5 | 81.6 | 81.8 | 73.7 | 84.9 |
| DINOv2 [31] | 768 | 65.9 | 60.0 | 80.5 | 79.7 | 71.1 | 82.1 |
| CroCo-v2 [44] | 768 | 65.5 | 67.0 | **87.0** | 81.5 | 73.8 | **89.4** |
| MaskDINO* [28] | 768 | 54.8 | 38.3 | 35.0 | 58.7 | 39.7 | 36.5 |
| LoCo (Ours) | 768 | **66.5** | **67.5** | 84.5 | **83.5** | **76.3** | 88.8 |
| LoCo (w/ DINOv2 backbone) | 768 | 61.1 | 58.5 | 80.4 | 80.4 | 71.0 | 83.8 |
| LoCo (w/ LoCUS architecture) | 64 | 10.5 | 7.3 | 11.2 | 58.9 | 47.3 | 58.1 |

tasks. However, when training CroCo, the encoder features are first used by a transformer-based decoder module before a loss function is applied. This means that there is no incentive for CroCo features to be location-consistent under a simple and interpretable cosine-similarity operation, and would require a more complex adapter to support similarity-based operations.

## 4.4 Scene-Stable Panoptic Segmentation

The task of scene-stable panoptic segmentation was originally introduced by Kloepfer *et al.* [26]. Given a set of images of a single scene, the goal is to create a segmentation mask for the objects in each of these images, where, crucially, different views of the same individual object are labelled consistently with the same identity.

Formally, for a set $\mathcal{I} = \{I_i\}$ of images of the same scene, a set $\mathcal{C} = \{0, \ldots, L-1\} = \mathcal{C}^{\text{st}} \cup \mathcal{C}^{\text{th}}$ of *semantic* classes that is split into 'stuff' (amorphous classes such as floor, walls, etc) and 'things' (clearly distinct objects) subsets $\mathcal{C}^{\text{st}}$ and $\mathcal{C}^{\text{th}}$ respectively. The latter are also split into a set $\mathcal{O} = \{0, \ldots, N-1\}$ of object instance IDs within the scene. The goal is to map each pixel $p_j$ to its semantic class $c_j$ if $c_j \in \mathcal{C}^{\text{st}}$, and to its scene-wide object instance $o_j$ if $c_j \in \mathcal{C}^{\text{th}}$. This is similar to the standard panoptic segmentation task [25] with the crucial difference that the object instance indices are consistent across different images of the same scene.

This task requires the algorithm to not just differentiate between multiple object instances of the same object class as in the conventional instance segmentation task, but also to recognise when different images show the same object, which requires a broader understanding of the scene geometry. The need to retain consistent object identities across different images is reminiscent of panoptic video segmentation [24]. However, in our task the images are unordered and have much larger viewpoint changes, so methods cannot rely on pixel tracking or optical flow to keep object identities consistent.

As is standard in self-supervised learning [5, 16, 26, 31], we train a linear probe to predict class labels from the feature vectors for every image patch.

### 4.4.1 Datasets.

Both the Matterport3D [6] and ScanNet [10] datasets provide 3D-mesh reconstructions of their constituent environments, segmented into individual objects. This allows us to generate ground-truth segmentation masks for scene-consistent object segmentation by finding the individual objects in each scene that a ray through a given pixel intersects. Scripts to generate this data will be included in our code release. For the Matterport3D dataset, we evaluate on 18 unseen scenes, and for ScanNet we evaluate on 21 unseen scenes selected to show a range of different types of environments.

### 4.4.2 Results.

We report our results on the Matterport3D dataset and on the ScanNet dataset in Section 4.4.1. We measure the scene-stable panoptic segmentation performance using three metrics, which we calculate for each object instance individually (for each object instance treating the segmentation masks as

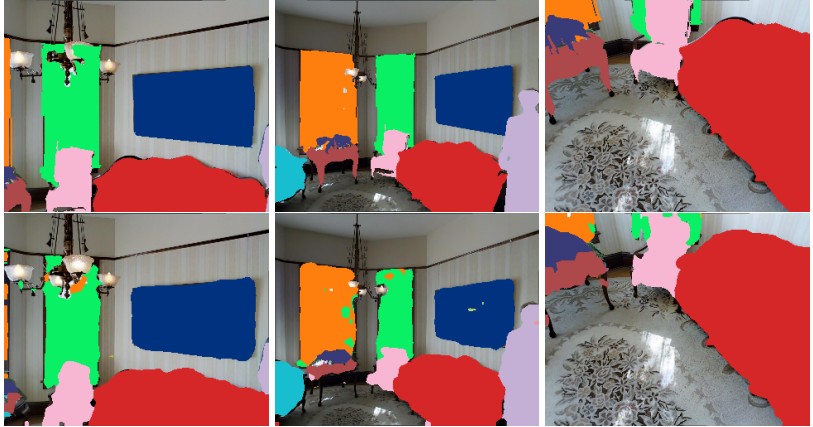

Figure 3: Scene-stable object segmentations for three images drawn from the Matterport3D [6] dataset. Ground-truth segmentations in the top row, predicted segmentations in the bottom row. The object identities and segmentation masks remain stable across significant viewpoint changes.

binary) and then average first across object instances in each scene and then across scenes. The Jaccard index (Jac) for each object instance is calculated by $TP/(FP + FN)$, given the counts for True Positive (TP), False Positive (FP), and False Negative (FN) predictions. The Intersection-over-Union (IoU) is the intersection-over-union with the ground-truth masks. The Average Precision (AP) is that of the linear classifier in a one-vs-all mode, taking all other pixels as negative labels.

Our LoCo-trained features perform better than all baselines, and have comparable performance to CroCo-v2 [44], which also makes use of multi-view supervision. However, ours has far fewer trainable parameters (only 28.9 million), since we only train a comparatively small CNN. In contrast, CroCo trains the entire network (85 million parameters) with a multi-view loss and on significantly larger datasets with greater computational resources (8 A100 GPUs *vs*. 1 RTX8000 GPU). We note also the strong performance of the original DINO compared to the newer DINOv2 method. As in the pixel correspondence task, this might be due to the final DINOv2 features focusing on semantic meaning, and so struggling to differentiate between, *e.g*., different chairs.

## 4.5 Ablations

**DINOv2 Backbone.** We also train our method using a frozen DINOv2 [31] backbone, again training a convolutional neural network to learn additive residuals and keeping other hyperparameters the same.

The resulting features significantly outperform the original pre-trained DINOv2 features for finding accurate pixel correspondences (Table 1), showing the advantage of LoCo-training in tasks that require location-consistent features. However, it slightly underperforms the LoCo model trained with the DINO-ViT-Base8 backbone. We hypothesize that this arises from the coarser feature map of the DINOv2-ViT-Base14 backbone (with a patch size of 14 instead of 8).

On the panoptic scene-stable segmentation task (Section 4.4.1), the LoCo model trained with the DINOv2 backbone only outperforms the original DINOv2 feature extractor on some of the metrics. This is likely attributable to the coarser feature map of this backbone, which leads to less fine-grained patch-level supervision during training.

**LoCUS Architecture.** To further investigate the impact of our alterations to the loss function and training algorithm compared to Kloepfer *et al*. [26], we train the original LoCUS architecture with the LoCo loss function and algorithm.

On the multi-view consistency task (Table 1), this model outperforms the original LoCUS model for small viewpoint changes, but underperforms for image pairs with larger viewpoint changes.

In fact, the LoCUS architecture trained with the LoCo-algorithm performs worse than the original LoCUS model on the panoptic scene-stable segmentation task (Section 4.4.1).

For this ablation, we trained for the same number of epochs as our other LoCo models, so it is possible that the vision transformer blocks in the LoCUS architecture require longer training times than the

convolutional layers of the LoCo models. In any case, this ablation illustrates that the improvements in memory efficiency do not by themselves lead to improvements in performance. Their advantage is that they allow for the training of larger models and higher-dimensional feature vectors with the same computational budget, the effect of which far outweighs any performance decrease due to our loss function and training algorithm changes.

**Effect of $\Delta$.** We analyse the impact of restricting further the range of similarities from which $\mathbb{N}_B^\alpha$ and $\mathbb{P}_B^\alpha$ are sampled by decreasing the saturation threshold $\Delta$. The results of are shown in the last two rows of Table 1. We see that while there is a small decrease in performance as the threshold decreases, overall, the training is remarkably robust to more aggressive filtering of patch pairs. This confirms the intuition that only a small number of patch pairs contribute meaningfully to the gradient and that most patch pairs can be discarded without significantly impacting the training behaviour.

### 4.6 Memory Efficiency Analysis

Using their training code, we find that the hyperparameters used in Kloepfer *et al.* [26] result in positive and negative pair set sizes of $|\mathbb{P}_B| \approx 13,000$ and $|\mathbb{N}_B| \approx 98,000$. The resulting matrix size, using single-precision floating point numbers, of $|\mathbb{P}_B| \times (|\mathbb{P}_B| + |\mathbb{N}_B|) \approx 51\text{GB}$ exceeds most computational limits, so the authors subsample $10\%$ of the negative pairs to reduce the matrix size to $5.7\text{GB}$ for a matrix with $1.4$ billion entries. We instead use a value of $|\mathbb{P}_B'| = 32$ and find empirically that even with the conservative value $\Delta$, roughly $80\%$ of the pair differences are well-approximated as having zero-gradient. This means that $|\mathbb{P}_B^\alpha| \approx (1 - 0.8)|\mathbb{P}_B|$ and $|\mathbb{N}_B^\alpha| \approx (1 - 0.8)|\mathbb{N}_B|$. Assuming the same computational budget of $5.7\text{GB}$, our method can therefore use batches $\mathbb{P}_B$ and $\mathbb{N}_B$ that are larger by a factor of $\sim 2000$. In our experiments, we limit $|\mathbb{P}_B^\alpha| < 800$ and $|\mathbb{N}_B^\alpha| < 3,000$, resulting in the matrix occupying only $500\text{KB}$ of memory, thereby freeing up GPU memory to train models with more parameters and with substantially larger feature vectors.

## 5 Conclusion

In this paper, we have proposed a method for the memory-efficient learning of location-consistent features. In particular, we present a reformulation of the smooth average precision ranking loss that corrects for biases induced by batching, and introduce an approximation that facilitates significant memory reductions without distorting the training signal. This mitigated a key memory bottleneck, allowing larger models to be trained with the same computational resources. Equipped with this novel retrieval-based objective function, we are able to efficiently learn to modulate DINO [5, 31] ViT features towards location-consistency.

Our feature extractor demonstrates compelling performance on the downstream tasks of scene-stable panoptic segmentation and visual place recognition, outperforming previous state-of-the-art feature extractors. This work goes some way towards scaling up the training pipeline; however, there is significant scope for applying these techniques on truly large scale image or video data in an entirely self-supervised manner by estimating depth maps and camera poses using off-the-shelf methods.

**Acknowledgements.** The authors acknowledge the generous support of the Royal Academy of Engineering (RF\201819\18\163), and EPSRC (VisualAI, EP/T028572/1).

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

Table 3: Hyperparameters of the convolutional layers of the residual network used for the pixel-correspondence task in Section 4.3.

| Layer | Kernel Size | Output Dimension | Dilation | Downsample Factor |
|-------|-------------|------------------|----------|-------------------|
| 1 | 5 | 64 | 1 | 2 |
| 2 | 5 | 128 | 1 | 2 |
| 3 | 5 | 256 | 1 | 2 |
| 4 | 5 | 512 | 2 | 1 |
| 5 | 5 | 768 | 2 | 1 |
| 6 | 5 | 768 | 2 | 1 |

## A  Architecture of Residual Network

In Table 3 we describe in more detail the architecture of our fully convolutional network that we use to compute residuals to frozen DINO [5] features. For the respective layers we downsample the resolution by the given factor using a BlurPool layer.

## B  Correction Factors for Batched Loss

### B.1  Derivation of Correction Factors

In the following we will derive the batch correction factors $|\mathbb{P}|/|\mathbb{P}_B|$ and $|\mathbb{N}|/|\mathbb{N}_B|$ from Sec. 3.4 in the main paper. We will also retain all notation conventions used in the main paper.

We can re-write the Eq. (2) as

$$\mathcal{L} = -\frac{1}{|\mathbb{P}|} \sum_{c_\alpha \in \mathbb{P}} \mathcal{L}_\alpha \tag{11}$$

$$\text{with } \mathcal{L}_\alpha = \frac{1 + \sum_{c_\beta \in \mathbb{P} \setminus \{c_\alpha\}} \sigma_\tau(s_\beta - s_\alpha)}{1 + \sum_{c_\beta \in \mathbb{P} \setminus \{c_\alpha\}} \sigma_\tau(s_\beta - s_\alpha) + \sum_{c_\gamma \in \mathbb{N}} \sigma_\tau(s_\gamma - s_\alpha)}.$$

and similarly Eq. (3) as

$$\mathcal{L}_B = -\frac{1}{|\mathbb{P}_B|} \sum_{c_\alpha \in \mathbb{P}} \mathcal{L}_{B\alpha} \tag{12}$$

$$\text{with } \mathcal{L}_{B\alpha} = \frac{1 + f_\mathbb{P} \sum_{c_\beta \in \mathbb{P}_B \setminus \{c_\alpha\}} \sigma_\tau(s_\beta - s_\alpha)}{1 + f_\mathbb{P} \sum_{c_\beta \in \mathbb{P}_B \setminus \{c_\alpha\}} \sigma_\tau(s_\beta - s_\alpha) + f_\mathbb{N} \sum_{c_\gamma \in \mathbb{N}_B} \sigma_\tau(s_\gamma - s_\alpha)},$$

where $f_\mathbb{P}$ and $f_\mathbb{N}$ are the as of yet unknown batch correction factors.

We want to choose $f_\mathbb{P}$ and $f_\mathbb{N}$ such that

$$\mathcal{L} = \mathbb{E}[\mathcal{L}_B]. \tag{13}$$

From the linearity of the expected value we have

$$\mathbb{E}[\mathcal{L}_B] = -\frac{1}{|\mathbb{P}_B|} \sum_{c_\alpha \in \mathbb{P}} \mathbb{E}[\mathcal{L}_{B\alpha}]. \tag{14}$$

Computing the expectation value of $\mathcal{L}_B$ exactly is challenging, as numerator and denominator are not independent. We can approximate $\mathbb{E}\mathcal{L}_{B\alpha}$ using the ratio estimator [37], which for two random variables $X$ and $Y$ approximates

$$\mathbb{E}\left[\frac{X}{Y}\right] \approx \frac{\mathbb{E}[X]}{\mathbb{E}[Y]}. \tag{15}$$

This estimator can be shown to be consistent, though biased. If the expectation values $\mathbb{E}[X]$ and $\mathbb{E}[Y]$ are estimated using $n$ samples, the bias can be shown to decrease as $O(n^{-1})$ [9].

Applying this estimator to Eq. (14) with

$$X = 1 + f_{\mathbb{P}} \cdot \sum_{c_\beta \in \mathbb{P}_B \setminus \{c_\alpha\}} \sigma_\tau(s_\beta - s_\alpha) \tag{16}$$

and

$$Y = 1 + f_{\mathbb{P}} \cdot \sum_{c_\beta \in \mathbb{P}_B \setminus \{c_\alpha\}} \sigma_\tau(s_\beta - s_\alpha) + f_{\mathbb{N}} \cdot \sum_{c_\gamma \in \mathbb{N}_B} \sigma_\tau(s_\gamma - s_\alpha), \tag{17}$$

we get

$$\mathbb{E}\left[\mathcal{L}_B\right] = \frac{1 + f_{\mathbb{P}} \cdot \mathbb{E}\left[\sum_{c_\beta \in \mathbb{P}_B \setminus \{c_\alpha\}} \sigma_\tau(s_\beta - s_\alpha)\right]}{1 + f_{\mathbb{P}} \cdot \mathbb{E}\left[\sum_{c_\beta \in \mathbb{P}_B \setminus \{c_\alpha\}} \sigma_\tau(s_\beta - s_\alpha)\right] + f_{\mathbb{N}} \cdot \mathbb{E}\left[\sum_{c_\gamma \in \mathbb{N}_B} \sigma_\tau(s_\gamma - s_\alpha)\right]}. \tag{18}$$

Since for uniform sampling of pairs from $\mathbb{P}$

$$\mathbb{E}\left[\sigma_\tau(s_\beta - s_\alpha)\right] = \frac{1}{|\mathbb{P}|} \sum_{c_\beta \in \mathbb{P} \setminus \{c_\alpha\}} \sigma_\tau(s_\beta - s_\alpha), \tag{19}$$

we obtain

$$\mathbb{E}\left[\sum_{c_\beta \in \mathbb{P}_B \setminus \{c_\alpha\}} \sigma_\tau(s_\beta - s_\alpha)\right] = \frac{|\mathbb{P}_B|}{|\mathbb{P}|} \sum_{c_\beta \in \mathbb{P} \setminus \{c_\alpha\}} \sigma_\tau(s_\beta - s_\alpha) \tag{20}$$

and similarly for $\mathbb{E}[\sum_{c_\beta \in \mathbb{N}_B} \sigma_\tau(s_\beta - s_\alpha)]$. Plugging this into Eq. (14) and Eq. (13), immediately results in

$$f_{\mathbb{P}} = \frac{|\mathbb{P}|}{|\mathbb{P}_B|} \tag{21}$$

$$f_{\mathbb{N}} = \frac{|\mathbb{N}|}{|\mathbb{N}_B|}. \tag{22}$$

We now investigate the bias due to the approximation Eq. (15). By Taylor expanding Eq. (15) around $X = \mathbb{E}[X] = \mu_X$ and $Y = \mathbb{E}[Y] = \mu_Y$ to second order we get

$$\frac{X}{Y} = \frac{\mu_X}{\mu_Y} + (X - \mu_X) \cdot \frac{1}{\mu_Y} - (Y - \mu_Y) \cdot \frac{\mu_x}{\mu_y^2} \tag{23}$$
$$+ \frac{1}{2}(X - \mu_X)^2 \cdot 0 + (Y - \mu_Y)^2 \cdot \frac{\mu_X}{\mu_Y^3}$$
$$- (X - \mu_X)(Y - \mu_Y) \cdot \frac{1}{\mu_Y^2} + \text{higher order terms.}$$

Applying the expectation value operator results in

$$\mathbb{E}\left[\frac{X}{Y}\right] = \frac{\mu_X}{\mu_Y} + \mathbb{E}\left[(Y - \mu_Y)^2\right] \cdot \frac{\mu_X}{\mu_Y^3} \tag{24}$$
$$- \mathbb{E}\left[(X - \mu_X)(Y - \mu_Y)\right] \cdot \frac{1}{\mu_Y^2} + \text{higher order terms.}$$

Using our definitions of $X$ and $Y$ from Eq. (16) and Eq. (17) we get

$$(X - \mu_X) = \frac{|\mathbb{P}|}{|\mathbb{P}_B|} \sum_{c_\beta \in \mathbb{P}_B \setminus \{c_\alpha\}} \sigma_\tau(s_\beta - s_\alpha) - |\mathbb{P}| \cdot \mathbb{E}_{c_\beta \in \mathbb{P} \setminus \{c_\alpha\}}[\sigma_\tau(s_\beta - s_\alpha)] \tag{25}$$

$$= O\left(\frac{1}{\sqrt{|\mathbb{P}_B|}}\right) \tag{26}$$

because the standard error of the sample mean decreases with the inverse square root of the number of samples.

Analogously, we have

$$(Y - \mu_Y) = O\left(\frac{1}{\sqrt{|\mathbb{P}_B|}}\right) + O\left(\frac{1}{\sqrt{|\mathbb{N}_B|}}\right) = O\left(\frac{1}{\sqrt{|\Omega_B|}}\right), \tag{27}$$

which then results in

$$\mathbb{E}\left[\frac{X}{Y}\right] = \frac{\mathbb{E}[X]}{\mathbb{E}[Y]} + O\left(\frac{1}{|\Omega_B|}\right). \tag{28}$$

The bias therefore decreases with the inverse of the batch size.

### B.2   Error of Prior Loss Functions

Previous uses of the Smooth Average Precision loss function in Brown *et al.* [4] and Kloepfer *et al.* [26] did not use batch correction factors.

The additional bias that results from this grows as the batch size decreases, because the batch correction factors decrease for larger batch sizes, and also because for larger batch sizes the sums in numerator and denominator grow and the additional $+1$ term that is unaffected by the batch correction factors becomes less and less relevant.

In our experiments, the sums would generally have magnitudes of at least $O(10^4)$, so the main function of the batch correction factors is to correctly balance the effect of positive and negative pairs in the denominator if $f_\mathbb{P} \neq f_\mathbb{N}$. This is the case in our experiments, where our hyperparameter choices mean that we generally sample a larger fraction of positive than of negative pairs so that $f_\mathbb{P}/f_\mathbb{N} \approx 0.1$. Without the batch correction factors, the loss function would place excessive weight on increasing the similarity of positive pairs.

Since these correction factors are simple to calculate, it is therefore valuable to use the loss function that includes them and that is built on a stronger theoretical foundation.

## C   Correction Terms for Memory Efficiency Improvements

### C.1   Derivation

We derive here the exact form of the correction term $\delta_\alpha^-$ in Eq. (9) of the main paper. The derivation for $\delta_\alpha^+$ proceeds entirely analogously.

Using the notation from the main paper, we can write

$$\frac{|\mathbb{N}|}{|\mathbb{N}_B|} \sum_{c_\beta \in \mathbb{N}_B} \sigma_\tau(s_\beta - s_\alpha) = \frac{|\mathbb{N}|}{|\mathbb{N}_B|} \sum_{c_\beta \in \underline{\mathbb{N}}_B^\alpha} \sigma_\tau(s_\beta - s_\alpha) \tag{29}$$

$$+ \frac{|\mathbb{N}|}{|\mathbb{N}_B|} \sum_{c_\beta \in \mathbb{N}_B^\alpha} \sigma_\tau(s_\beta - s_\alpha)$$

$$+ \frac{|\mathbb{N}|}{|\mathbb{N}_B|} \sum_{c_\beta \in \overline{\mathbb{N}}_B^\alpha} \sigma_\tau(s_\beta - s_\alpha).$$

Using the approximation from Eq. (4) in the main paper, we have

$$\forall c_\beta \in \underline{\mathbb{N}}_B^\alpha : \sigma_\tau(s_\beta - s_\alpha) \approx 0 \tag{30}$$

and

$$\forall c_\beta \in \overline{\mathbb{N}}_B^\alpha : \sigma_\tau(s_\beta - s_\alpha) \approx 1. \tag{31}$$

Plugging into Eq. (29) yields

$$\frac{|\mathbb{N}|}{|\mathbb{N}_B|} \sum_{c_\beta \in \mathbb{N}_B} \sigma_\tau(s_\beta - s_\alpha) \approx \frac{|\mathbb{N}|}{|\mathbb{N}_B|} \sum_{c_\beta \in \mathbb{N}_B^\alpha} \sigma_\tau(s_\beta - s_\alpha) + \frac{|\mathbb{N}|}{|\mathbb{N}_B|} \cdot |\overline{\mathbb{N}}_B^\alpha|, \tag{32}$$

with the second term exactly the correction term $\delta_\alpha^- = |\overline{\mathbb{N}}_B^\alpha| \frac{|\mathbb{N}|}{|\mathbb{N}_B|}$.

## C.2 Approximation Error

Using the sum expansion of Eq. (29) in Eq. (12) results in

$$\mathcal{L}_{B\alpha} = \left(1 + \frac{|\mathbb{P}|}{|\mathbb{P}_B|} \sum_{c_\beta \in \underline{\mathbb{P}}_B^\alpha} \sigma_\tau(s_\beta - s_\alpha)\right. \tag{33}$$

$$+ \frac{|\mathbb{P}|}{|\mathbb{P}_B|} \sum_{c_\beta \in \mathbb{P}_B^\alpha \setminus \{c_\alpha\}} \sigma_\tau(s_\beta - s_\alpha)$$

$$\left. + \frac{|\mathbb{P}|}{|\mathbb{P}_B|} \sum_{c_\beta \in \overline{\mathbb{P}}_B^\alpha} \sigma_\tau(s_\beta - s_\alpha)\right)$$

$$\left/ \left(1 + \frac{|\mathbb{P}|}{|\mathbb{P}_B|} \sum_{c_\beta \in \underline{\mathbb{P}}_B^\alpha} \sigma_\tau(s_\beta - s_\alpha)\right.\right.$$

$$+ \frac{|\mathbb{P}|}{|\mathbb{P}_B|} \sum_{c_\beta \in \mathbb{P}_B^\alpha \setminus \{c_\alpha\}} \sigma_\tau(s_\beta - s_\alpha)$$

$$+ \frac{|\mathbb{P}|}{|\mathbb{P}_B|} \sum_{c_\beta \in \overline{\mathbb{P}}_B^\alpha} \sigma_\tau(s_\beta - s_\alpha)$$

$$+ \frac{|\mathbb{N}|}{|\mathbb{N}_B|} \sum_{c_\beta \in \underline{\mathbb{N}}_B^\alpha} \sigma_\tau(s_\beta - s_\alpha)$$

$$+ \frac{|\mathbb{N}|}{|\mathbb{N}_B|} \sum_{c_\beta \in \mathbb{N}_B^\alpha} \sigma_\tau(s_\beta - s_\alpha)$$

$$\left.\left. + \frac{|\mathbb{N}|}{|\mathbb{N}_B|} \sum_{c_\beta \in \overline{\mathbb{N}}_B^\alpha} \sigma_\tau(s_\beta - s_\alpha)\right)\right.$$

In the worst case, the inequalities defining $\underline{\mathbb{N}}_B^\alpha, \overline{\mathbb{N}}_B^\alpha, \underline{\mathbb{P}}_B^\alpha, \overline{\mathbb{P}}_B^\alpha$ are tight, so

$$\forall c_\beta \in \underline{\mathbb{N}}_B^\alpha \cup \underline{\mathbb{P}}_B^\alpha : \sigma_\tau(s_\beta - s_\alpha) = -\Delta \tag{34}$$

$$\forall c_\beta \in \overline{\mathbb{N}}_B^\alpha \cup \overline{\mathbb{P}}_B^\alpha : \sigma_\tau(s_\beta - s_\alpha) = \Delta. \tag{35}$$

Plugging in and using $\sigma_\tau(\Delta) = 1 - \sigma_\tau(-\Delta)$ yields

$$
\mathcal{L}_{B\alpha} = \left( 1 + \frac{|\mathbb{P}|}{|\mathbb{P}_B|} |\mathbb{P}_B^\alpha| \sigma_\tau(-\Delta) \right. \tag{36}
$$

$$
+ \frac{|\mathbb{P}|}{|\mathbb{P}_B|} \sum_{c_\beta \in \mathbb{P}_B^\alpha \setminus \{c_\alpha\}} \sigma_\tau(s_\beta - s_\alpha)
$$

$$
\left. + \delta_\alpha^+ (1 - \sigma_\tau(-\Delta)) \right)
$$

$$
\bigg/ \left( 1 + \frac{|\mathbb{P}|}{|\mathbb{P}_B|} |\mathbb{P}_B^\alpha| \sigma_\tau(-\Delta) \right. \tag{37}
$$

$$
+ \frac{|\mathbb{P}|}{|\mathbb{P}_B|} \sum_{c_\beta \in \mathbb{P}_B^\alpha \setminus \{c_\alpha\}} \sigma_\tau(s_\beta - s_\alpha)
$$

$$
+ \delta_\alpha^+ (1 - \sigma_\tau(-\Delta))
$$

$$
+ \frac{|\mathbb{N}|}{|\mathbb{N}_B|} |\mathbb{N}_B^\alpha| \sigma_\tau(-\Delta)
$$

$$
+ \frac{|\mathbb{N}|}{|\mathbb{N}_B|} \sum_{c_\beta \in \mathbb{N}_B^\alpha} \sigma_\tau(s_\beta - s_\alpha)
$$

$$
\left. + \delta_\alpha^- (1 - \sigma_\tau(-\Delta)) \right).
$$

Taking a Taylor expansion with respect to $z \equiv \sigma_\tau(-\Delta) \approx 0$ around $z = 0$ to first order gives

$$
\mathcal{L}_{B\alpha} = \frac{1 + \frac{|\mathbb{P}|}{|\mathbb{P}_B|} \sum_{c_\beta \in \mathbb{P}_B^\alpha \setminus \{c_\alpha\}} \sigma_\tau(s_\beta - s_\alpha) + \delta_\alpha^+}{1 + \frac{|\mathbb{P}|}{|\mathbb{P}_B|} \sum_{c_\beta \in \mathbb{P}_B^\alpha \setminus \{c_\alpha\}} \sigma_\tau(s_\beta - s_\alpha) + \frac{|\mathbb{N}|}{|\mathbb{N}_B|} \sum_{c_\beta \in \mathbb{N}_B^\alpha} \sigma_\tau(s_\beta - s_\alpha) + \delta_\alpha^+ + \delta_\alpha^-} \tag{38}
$$

$$
+ z \cdot \left[ \frac{|\mathbb{N}||\mathbb{P}|}{|\mathbb{N}_B||\mathbb{P}_B|} (|\mathbb{P}_B^\alpha| - |\overline{\mathbb{P}}_B^\alpha|) \sum_{c_\beta \in \mathbb{N}_B^\alpha} \sigma_\tau(s_\beta - s_\alpha) \right.
$$

$$
+ \delta_\alpha^- \left( 1 + \frac{|\mathbb{P}|}{|\mathbb{P}_B|} |\mathbb{P}_B^\alpha| + \frac{|\mathbb{P}|}{|\mathbb{P}_B|} \sum_{c_\beta \in \mathbb{P}_B^\alpha \setminus \{c_\alpha\}} \sigma_\tau(s_\beta - s_\alpha) \right)
$$

$$
\left. - \frac{|\mathbb{N}|}{|\mathbb{N}_B|} |\mathbb{N}_B^\alpha| \left( 1 + \delta_\alpha^+ + \frac{|\mathbb{P}|}{|\mathbb{P}_B|} \sum_{c_\beta \in \mathbb{P}_B^\alpha \setminus \{c_\alpha\}} \sigma_\tau(s_\beta - s_\alpha) \right) \right]
$$

$$
\bigg/ \left( 1 + \frac{|\mathbb{P}|}{|\mathbb{P}_B|} \sum_{c_\beta \in \mathbb{P}_B^\alpha \setminus \{c_\alpha\}} \sigma_\tau(s_\beta - s_\alpha) + \frac{|\mathbb{N}|}{|\mathbb{N}_B|} \sum_{c_\beta \in \mathbb{N}_B^\alpha} \sigma_\tau(s_\beta - s_\alpha) + \delta_\alpha^+ + \delta_\alpha^- \right)^2
$$

$$
+ O(z^2)
$$

We see that the loss function used in the main paper (Eq. (9)) re-appears as the zeroth order term of this expansion, while the higher order term(s) provide an upper bound for the error due to approximating the sigmoids in the worst case.

## D  Implementation Details for Efficient Patch-Pair Sampling

We pre-compute the 3D world coordinates of each patch in the dataset by backprojecting the centre coordinate into the scene using information about camera pose and depth values at that patch.

During training, we sample the positive patch-pairs $c_\alpha \in \mathbb{P}$ by uniformly randomly sampling a patch from a scene and then using the highly efficient FAISS [11] library to find and sample a patch within the positive region of the original patch. This approach does not uniformly sample from all positive

patch-pairs since patches that are part of a larger number of positive pairs are not more likely to be chosen, but computing the number of positive pairs that include a given patch to correct for this would be computationally unfeasible. This also does not change our derivations, since all pairs are still sampled from the same (even though non-uniform) distribution.

In each training step, we extract feature maps for only those images that contain the patches for the $c_\alpha$ pairs. This is to use our limited computational budget in a way that maximises $|\mathbb{P}'_B|$. The pairs $c_\beta \in \mathbb{P}$ and $c_\gamma \in \mathbb{N}$ are then sampled from among the patches in the images for which feature maps have been extracted. This samples using the same distribution as the $c_\alpha$ pairs, since for each $c_\alpha$ the other images in the batch represent merely a uniformly random restriction of the patches that can be sampled.

# E    Comparison with Hard Negative Mining

## E.1    Discussion

One of the contributions of this paper, selecting pairs of patches that do not saturate a sigmoid function in the loss function (*cf*. sec. 3.5), is reminiscent of the better-known technique of hard negative mining in self-supervised learning [21, 22, 33].

However, there are three important differences:

Firstly, our method selects pairs of samples rather than individual hard (negative) samples, a result of our general reformulation of the problem of learning location-consistent features. In hard negative mining, given a *single* anchor sample, other samples are selected based on their *features*, whereas in our case, given a *pair* of samples (patches), other *pairs* of patches are selected based on their *similarity*.

Secondly, while hard negative mining is a one-sided filter, our method is two-sided. We do not just filter out 'too easy' pairs like hard negative mining, *i.e.* pairs that are easily and correctly classified into positive or negative pairs, but also 'too difficult' pairs, *i.e.* pairs that are confidently but incorrectly classified. Both kinds of sample pairs saturate the sigmoid function and do not contribute to the gradient. Most standard contrastive learning loss functions do not contain a sigmoid function, so such a two-sided filter is not appropriate there.

Thirdly, hard negative mining approaches aim to improve downstream model performance by placing more weight on difficult samples. The hard negative sampling changes the loss function to learn more useful features. In contrast, our method aims exclusively to reduce the training algorithm's memory consumption without impacting the loss landscape.

## E.2    Performance of NCE with Hard Sample Mining

We also trained a model using a more standard contrastive learning loss together with hard mining. We again use a modified version of the model architecture of DINO-Tracker [41], training a fully convolutional neural network to learn residual features to frozen DINO [5] features.

Instead of our loss, we train using the noise contrastive estimation (NCE) loss function [14]. From a batch of 24 images, we sample 256 query patches, for each of which we sample 128 positive and 384 negative patches.

We use hard sample mining for both positive and negative samples. Inspired by Robinson *et al.* [33], we sample the negative samples randomly from among the patches in the negative region with a probability $\propto \exp(\beta s_{ij})$, where $s_{ij}$ is the similarity between the query patch and the potential negative patch. The coefficient $\beta$ is a hyperparameter that we set to $\beta = 0.1$. Similarly, we sample the positive patches from among the patches in the positive region with a probability $\propto \exp(-\beta s_{ij})$. This means that we over-sample negative patches with high similarities and positive patches with low similarities, as these are the worst-performing samples.

The results of this model on the pixel correspondence estimation task are shown in Table 4. Though it outperforms the baseline models and does comparatively well at large viewpoint changes, our loss is more suitable for learning multi-view consistent features.

Table 4: Results on the pixel correspondence task on the Paired split [36] of ScanNet [10], as introduced by El Banani *et al*. [12]. We report the recall of accurate pixel correspondences at a reprojection error threshold of 10 pixels, for image pairs with the respective viewpoint changes.

| Model | $0°-15°$ | $15°-30°$ | $30°-60°$ | $60°-180°$ |
|---|---|---|---|---|
| LoCUS† [26] | 23.5 | 18.8 | 13.5 | 7.5 |
| DINO [5] | 45.0 | 34.3 | 22.6 | 10.7 |
| DINOv2 [31] | 37.0 | 27.5 | 19.7 | 11.2 |
| DINOv2 [31], Blocks 1–6 | 47.1 | 36.4 | 22.4 | 8.4 |
| CroCo-v2 [44] | 16.8 | 12.4 | 7.4 | 3.7 |
| LoCo (Ours) | 61.8 | 52.7 | 31.8 | 10.3 |
| NCE + Hard Mining | 50.6 | 40.8 | 26.7 | 11.4 |

# F    Visual Place Recognition

We also demonstrate the utility of location-consistent features on the task of Visual Place Recognition (VPR). Given a set $\mathcal{R}$ of reference images and a set $\mathcal{Q}$ of query images with $\mathcal{Q} \cap \mathcal{R} = \emptyset$, a ground truth function $g : \mathcal{Q} \to 2^{\mathcal{R}}$ maps a query image to the subset of reference images that show the same 'place' as the query image. Since the reference set is typically large, here we focus on global descriptor methods, which compute a global descriptor for each image and for each query image retrieve the most similar reference images.

Recently, Keetha *et al*. [23] demonstrated that excellent place recognition performance can be achieved using a simple recipe: extract patch-level feature maps using a pre-trained feature extractor, then aggregate these into a global image descriptor using the VLAD [20] algorithm. We follow this approach and report the results for different feature extractors. We use 128 clusters for VLAD and construct the VLAD vocabulary only from the respective set of reference images.

## F.1    Datasets.

The indoor datasets that are typically used to evaluate VPR methods (*e.g.* Baidu Mall [38], Gardens Point [39], and 17Places [35]) tend to suffer from one of two defects. For some, like Gardens Point, the development of large-scale pre-trained image models means that VPR performance is strong enough that these datasets can no longer resolve small performance differences, with Keetha *et al*. [23] reporting 99.5% Recall@5 for three variants of their method. Others, like Baidu Mall and 17Places, suffer from poor ground truth. Reference images labelled as 'ground truth' frequently have no image overlap with the query image or do *not* include images with significant overlap as ground truth. We therefore use the pose and depth information available with the Matterport3D and ScanNet datasets to construct VPR datasets for which the IoU between query and ground-truth images falls between $0.2$ and $0.4$. This ensures that the place recognition task is challenging while remaining tractable for purely vision-based systems. The code to deterministically generate the datasets will be released publicly.

The images that comprise these datasets are drawn from environments that were unseen during training. For Matterport3D, using images from 18 unseen scenes results in 8637 reference and 2194 query images, and for ScanNet, using images from 21 unseen scenes results in 4295 reference and 239 query images.

## F.2    Results.

We give Visual Place Recognition results in Table 5, quoting the Recall@1 and Recall@5. While the performance of our LoCo features remains respectable, it does not outperform the baseline feature extractors by as much as in the scene-stable panoptic segmentation task. This is potentially due to the VLAD feature aggregation discarding information about the relative arrangement of different feature vectors in the image, thereby removing a significant advantage of location-stable features compared to feature extractors focused more on semantic information like DINO.

Table 5: Visual Place Recognition Results on VPR datasets constructed from unseen images in Matterport3D [6] and ScanNet [10]. MixVPR produces global image descriptors directly, all other methods extract $d$-dimensional feature vectors for $30 \times 40$ patches that are then aggregated into global descriptors using VLAD. LoCUS$^{\dagger}$ [26] uses $d = 64$, rather than $d = 768$.

| Model | Matterport3D | | ScanNet | |
| | R@1 | R@5 | R@1 | R@5 |
|---|---|---|---|---|
| LoCUS$^{\dagger}$ [26] | 20.8 | 38.4 | 61.1 | 79.9 |
| DINO [5] | 45.7 | 77.8 | 81.0 | **92.9** |
| DINOv2 [31] | 25.0 | 52.0 | 59.4 | 78.2 |
| CroCo-v2 [44] | 26.6 | 51.6 | 63.6 | 81.1 |
| MixVPR [1] | 26.9 | 51.4 | 72.4 | 84.5 |
| LoCo (Ours) | **46.8** | **79.1** | **81.2** | 92.5 |

## G   Qualitative Visualisations

In Fig. 4, we show point-cloud reconstructions obtained by projecting the pixel patches of a set of images showing a living room into 3D, and colouring the each point by the similarity of its patch's feature vector with a query feature vector taken from a query patch. The query patch looks at a point on the wall to the top left of the picture hanging on the wall. The upper image shows the similarities as reconstructed using our LoCo method, the lower as reconstructed using DINO [5] features.

The region of space where feature vectors have a high similarity with the query feature is much more localised for our method than for the DINO features.

## H   Limitations

The training algorithm described in the paper assumes the availability of camera poses and depth maps for each RGB image. While a number of datasets exist for whom this information is available, this may not always be the case. There exist a number of comparatively accurate algorithms to estimate this information, but the paper does not investigate whether such methods are sufficiently accurate to reproduce the stated performance.

The paper also limits itself to training comparatively small convolutional networks, or finetuning only a small number of layers of a pre-trained foundation model rather than training a similarly sized model from scratch. There is therefore the possibility that the training algorithm described requires an initialisation with relatively high-quality features and is not entirely appropriate for training visual foundation models from scratch.

Thirdly, the paper only runs experiments training on a comparatively small dataset only containing a few dozen different environments. While all testing is performed on unseen environments, the paper does not investigate the existence of scaling laws as available training data increases.

## I   Broader impacts

This work, on memory-efficient learning of 3D-consistent image features, has potential positive and negative social impacts. On the positive side, reducing the memory and compute requirements of a common type of self-supervised loss (a ranking loss) permits a wider pool of researchers and interest groups to investigate and train models that use such losses, on restricted compute budgets. In addition, self-supervised learning reduces the need for often exploitative human labour practices associated with obtaining cheap labelled data.

On the negative side, training large self-supervised models has considerable environmental impacts due to carbon emissions that lead to significant societal impacts, including direct consequences (displacement, damage from extreme weather events, *etc*.) and indirect consequences. Moreover, location-consistent image features are likely to be useful for tracking, which can be invasive of privacy and can be used for malicious purposes.

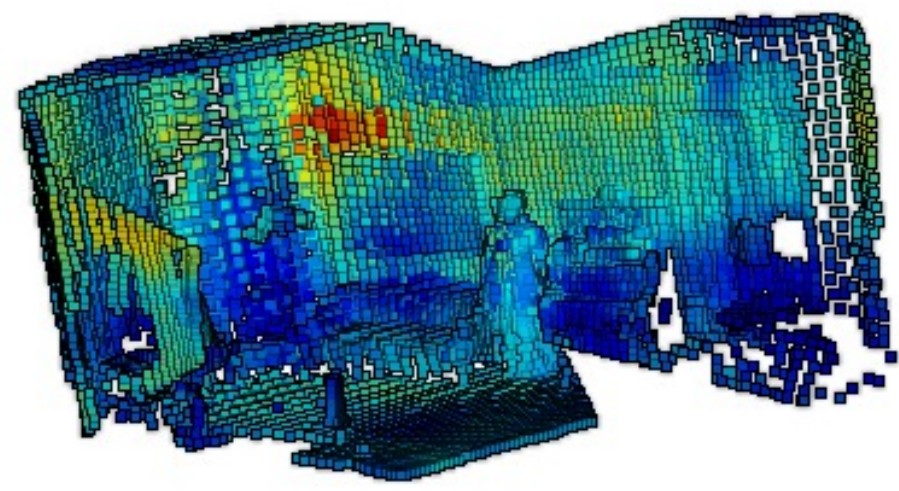

(a) Similarities of the features of patches taken from a set of images with a query feature vector, using our LoCo features, projected into 3D. The query feature vector's location is to the top left of the picture on the wall.

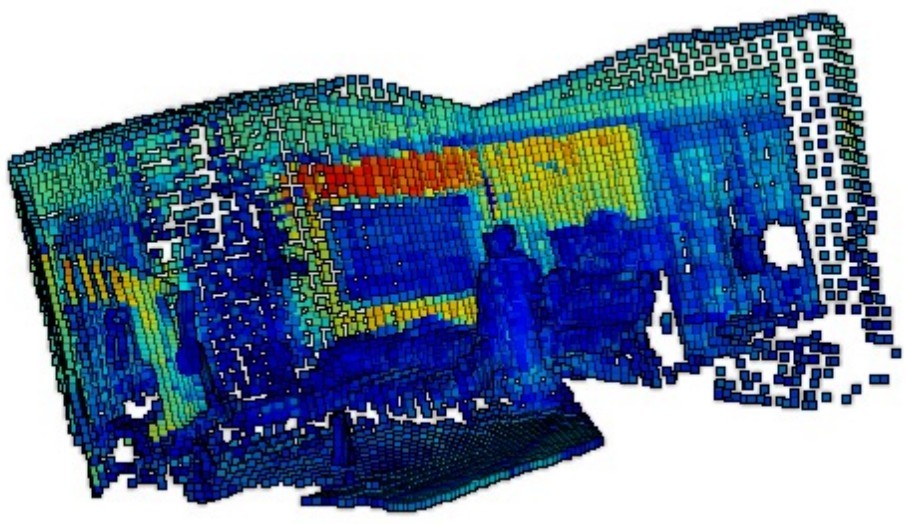

(b) Similarities of the features of patches taken from a set of images with a query feature vector, using DINO [5] features, projected into 3D. The query feature vector's location is to the top left of the picture on the wall.

Figure 4: 3D visualisation coloured by the similarity with respect to a query feature vector.

