# OpenReview forum: "LoCo: Learning 3D Location-Consistent Image Features with a Memory-Efficient Ranking Loss"
_NeurIPS.cc/2024/Conference — NeurIPS 2024 poster_

### Official Review · Reviewer_aW6W · 2024-07-11

**Soundness:** 3
**Presentation:** 4
**Contribution:** 3
**Rating:** 6
**Confidence:** 4

**Summary:**

This work focuses on the best strategy to pre-train a feature extractor network optimized to be invariant wrt to the viewpoint in the image. To do so the authors use a dataset of paired views from which they extract pairs of positive patches (same 3D point in two different views) and negatives (different 3D points). Given these sets a feature extractor model is trained for patch retrieval using a loss proposed by the authors. In particular the paper starts from the existing Average PRecision Loss and adapts it to make it significantly more memory efficient. Provided with the data and the loss the authors train a tiny CNN adapter on top of Dino features to make them more robust wrt to the viewpoint. The performance of the trained model is evaluated on few ad-hoc benchmarks (multi view feature consistency, consistency of panoptic segmentation and visual place retrieval) where the proposal achieves improvements wrt to the competitors.

**Strengths:**

+ Presentation: the paper is very well written and presented. The paper has a straightforward core idea and contribution and the author make a good job at motivating and explaining it.

+ Reusable: the proposed improved ranking loss can likelly be re-used for other tasks that require some form of metric learning and it’s not specifically tailored for the multi-view consistency use case. As such the contribution of this work can probably extend to nearby fields.

+ Efficiency: the proposal seems to be quite effective even when training somewhat shallow models on somewhat small datasets (although what the author train is a residual model on top of a pre-trained Dino one). This is quite interesting as it shows that a relatively small learning budget is enough for learning multi-view consistency.

**Weaknesses:**

a. Limited experimental validation: while the author tested their model against competitors across 3 tasks, these are mostly internal comparisons on tasks that would privilege the proposed model. In particular, Table 1 where the method has the biggest gains, is basically measuring the performance on the task LoCo was trained on vs models that were not trained for the task (except for LoCUS). Regarding LoCUS in table 1 it is shown with a † that has no match in the caption of the table, following the rest of the paper I’m assuming it means that the method only has dimensionality 64 for features.  It would have been interesting to include in the evaluation some of the correspondence based tasks that CroCo-v2 has been exploring like stereo depth estimation, optical flow or relative pose estimation.

b. Generalization: the model is trained on samples from matterport3D and evaluated on unseen scenes from Matterport and on scenes from ScanNet. This constrains quite a lot the scenarios being tested and does not give insight on the generalization capabilities of the proposed method. For example how would it perform in very different scenarios like an outdoor scene? Note that there are plenty of  dataset for evaluation of correspondences method that could be used for this task like [MegaDepth](https://www.cs.cornell.edu/projects/megadepth/)

c. Potential unfair comparison to competitors: the closest competitor that has been trained with a comparable objective to the one proposed by the authors is LoCUS that according to the evaluation in Tab.1 and Tab. 2 achieves consistently worse results. However LoCUS uses only 2 FC layers on top of the features f a frozen DINO (500K parameters) while the proposed methods use a full CNN (~20M parameters), as such the comparison is not particularly fair. It would have been interesting to have a version of the LoCo method trained in the exact same settings as LoCUS (models and data) to verify what is the gain or losses of the proposed new loss function compared to the one introduced in LoCUS.

**Questions:**

**Questions**


1. Can you comment on weakness [b] and whether generalization  is to be expected or very limited?

2. Can you comment on weakness [c]

**Typos**

* L 166: Eq. 12 → Eq. 3?

**Limitations:**

Limitations have been adequatelly discussed in the manuscript.

---

> ### Author Rebuttal · Authors · 2024-08-06
>
> Thank you for your detailed and constructive review. We appreciate your positive remarks regarding the presentation, efficiency, and potential reusability of our proposed method. We also acknowledge the valuable concerns you’ve raised and address them below.
>
> ---
>
> **Weakness (a): Limited experimental validation**
>
> We understand your concern regarding the focus of our experimental validation on tasks where our model excels. These tasks were chosen to highlight the core strengths of our approach in a controlled setting.
>
> For tasks like surface normal prediction and monocular depth estimation (as reported in, e.g., El Banani et al. [12]), the correct prediction depends not just on the scene geometry, but also on the camera pose. Our work however explicitly aims to make the features location-consistent and thereby invariant to changes in camera pose. The LoCo-trained features therefore lack crucial information for such tasks.
> In their experiments on optical flow and stereo depth estimation, CroCov2 [44] fine-tune their entire (pre-trained) network rather than training a comparatively small probe. Experiments of that scale were beyond our computational resources.
>
> In the appendix, we provide an evaluation on Visual Place Recognition, which further demonstrates the versatility and utility of the features learned by our method across different applications.
>
> ---
>
> **Weakness (b): Generalization capabilities**
>
> We appreciate the concern regarding generalization and agree that this is an important aspect of any method’s robustness. Due to computational limitations, we were constrained to training relatively small models on indoor scene datasets. In performing well on many unseen diverse indoor environments, the models do demonstrate the capability to generalise to unfamiliar environments. Due to the circumscribed training domain, we do not expect these models to perform well on environments with a larger domain shift (such as outdoor environments), where performance would mostly depend on the degree of their similarity with the training domain.
>
> However, the training method we propose is designed to be fully general and adaptable to a variety of settings. While our current experiments focus on indoor scenes, the methodology itself should be applicable to other scenarios and domains without significant adjustments.
>
> ---
>
> **Weakness (c): Potential unfair comparison to competitors**
>
> We appreciate your concern about the comparison between our method and LoCUS, given the differences in model architecture and parameter count. To address this, we conducted additional experiments, as detailed in our global author response, where we fine-tuned the LoCUS architecture using our memory-efficient loss.
>
> These experiments reveal that while our method does not outperform LoCUS in all tasks in terms of accuracy, it significantly enhances memory efficiency. This efficiency is what unlocks the training of larger models that produce higher-dimensional feature vectors within the same computational budget, leading to substantial overall performance improvements. The ability to scale up model capacity while maintaining computational feasibility is a key advantage of our approach, illustrating its potential beyond the specific settings and model scales we tested. We do however appreciate the importance of having this apples-to-apples comparison in the paper.
>
> ---
>
> **Typos**
>
> We will correct the typo on Line 166 (Eq. 12 → Eq. 3) and address the missing match for the dagger symbol in Table 1 before publication. We appreciate your attention to detail in pointing these out.
>
> ---
>
> We hope these clarifications address your concerns and provide additional context for evaluating our work. We are confident that our paper offers valuable contributions, particularly in advancing methods for training location-consistent feature extractors in a memory-efficient manner.
>
> We appreciate your careful consideration of our work and look forward to any further feedback.

---

> > ### Comment · Reviewer_aW6W · 2024-08-12
> > **Acknowledgement**
> >
> > Thank you authors for submitting a detailed rebuttal.
> > I would suggest to incorporate the result of LocUS (w Loco architecture) in the future versions of the work because they help quantifying the effect on performance of the tradeoff introduced by the memory optimized loss function used.

---

> > > ### Author Response · Authors · 2024-08-13
> > >
> > > Thank you - we will definitely include this result in our revision and agree that it helps more concretely pinpoint the advantages of the proposed approach.

---

### Official Review · Reviewer_YTBi · 2024-07-12

**Soundness:** 3
**Presentation:** 3
**Contribution:** 4
**Rating:** 6
**Confidence:** 2

**Summary:**

The paper introduces a memory efficient loss for location consistent image features. The paper addresses the problem of high memory footprint of previous work, by significantly reducing the memory footprint, by 3 orders of magnitude. Memory efficiency is achieved by sampling the positive pairs from a smaller subset and using a threshold on similarity differences between pairs, and discarding the pairs with gradients close to 0, which do not contribute to the loss. To compensate for the non-uniform sampling, some correction terms are added to the loss.

**Strengths:**

* Significant memory footprint reduction (3 orders of magnitude), allowing for larger batch sizes
* Identifying that the gradients from most pairs are close to 0 and don't contribute to the losses.
* Analysis of correction to the batched loss function.

**Weaknesses:**

* relies on existing 3D mesh reconstruction of the environment (for segmentation masks of individual objects in the scenes) -- this limits evaluation to datasets with existing mesh reconstructions.
* while direct comparison with LoCUS on the same feature dimension is not possible due to high memory consumption, it would be interesting to see a comparion with LoCo on a feature dimension d=64 (Tab 2) for a fair comparison.

**Questions:**

How does LoCo perform with features coming from different models (e.g. DINO vs DINO v2)?

**Limitations:**

- limited to datasets with 3D reconstructions available.

---

> ### Author Rebuttal · Authors · 2024-08-05
>
> Thank you for your thoughtful and encouraging review. We appreciate your recognition of the strengths of our work, particularly in terms of the memory efficiency gains and the analysis of the correction to the loss function. Below, we address your concerns and questions in more detail.
>
> ---
>
> **Weakness 1: Reliance on existing 3D mesh reconstruction for evaluation**
>
> We understand your concern regarding the reliance on 3D mesh reconstructions for the Scene-Stable Panoptic Segmentation task. However, we would like to clarify that this requirement is only necessary for *evaluation* purposes, not for training or general application of the LoCo features. The primary goal of this task is to highlight the improved location-consistency of our trained LoCo features compared to existing feature extractors.
>
> While it is true that this evaluation necessitates datasets with 3D reconstructions, we believe that this is not a significant limitation. Several existing datasets provide this data, and our method's ability to perform well in this setting demonstrates its robustness and applicability in scenarios requiring high-precision spatial understanding. The improvements in location-consistency, as demonstrated in this task, are transferable to other tasks that do not necessarily require 3D meshes.
>
> ---
>
> **Weakness 2: Comparison with LoCo at a feature dimension of d=64**
>
> We appreciate your suggestion to include a comparison between LoCo and LoCUS at a feature dimension of d=64 for a more direct comparison. In the global author response, we provided additional results for fine-tuning the LoCUS architecture using our memory-efficient loss. While it is true that a direct comparison at a high feature dimension is challenging due to memory constraints, our additional experiments demonstrate that even at a low feature dimension our method offers significant improvements in memory efficiency while maintaining strong performance.
>
> The improvements seen with larger models trained using our method further validate the value of our modifications to the loss function and training algorithm in overcoming memory limitations. This supports the scalability and robustness of our approach across different architectures and feature dimensions.
>
> ---
>
> **Question: Performance of LoCo with features from different models (e.g., DINO vs. DINOv2)**
>
> In response to your question about the performance of LoCo with features from different models, we have provided additional results using a DINOv2 ViT-Base backbone in the global author response. These results show that our method significantly outperforms the original DINOv2 feature extractor in tasks requiring accurate pixel correspondences.
>
> While the performance with the DINOv2 backbone is slightly lower than with the DINOv1 backbone, this likely arises from the differences in the patch size of the backbone, which affects the granularity of the feature map. Nevertheless, the fact that training with our method improves multi-view consistency with different backbone architectures demonstrates its flexibility and effectiveness across various feature extraction models.
>
> ---
>
> **Limitations: Dataset requirements**
>
> We acknowledge the limitation regarding the availability of 3D reconstructions in certain datasets. As discussed, this requirement is specific to the evaluation of certain tasks and not an inherent limitation of our method. The technique we propose is broadly applicable and can be adapted to other datasets and tasks that do not require 3D reconstruction data.
>
> ---
>
> We hope these clarifications address your concerns and provide a deeper understanding of the contributions and versatility of our work. We appreciate your constructive feedback and believe that our paper makes a valuable contribution to the field of vision foundation models, particularly in improving memory efficiency and multi-view consistency.
>
> Thank you again for your careful review and for the opportunity to improve our work through your insights.

---

> > ### Comment · Reviewer_YTBi · 2024-08-12
> >
> > I would like to thank the authors for their responses and the other reviewers for their thoughtful feedback and questions.
> >
> > I think the motivation of choice of Matterport (raised by Reviewer jhZZ ) and apples to apples comparison with LocUS (raised by Reviewer aW6W) should make it to the main paper, emphasizing the reduced memory footprint of the proposed method.

---

> > > ### Author Response · Authors · 2024-08-13
> > >
> > > Thank you, we will include these in the main paper to better reflect the setting and contributions of our proposed approach.

---

### Official Review · Reviewer_jhZZ · 2024-07-13

**Soundness:** 2
**Presentation:** 2
**Contribution:** 1
**Rating:** 3
**Confidence:** 5

**Summary:**

This paper presents a new training scheme for vision foundation models. The key goal of the paper is to enhance the multi-view consistency of vision foundation models. To this end, the paper revisits the idea from soft average precision, and applies the idea for training vision foundation models. Specifically, the loss loosely enforces similar score values for positive patch pairs, unlike explicitly enforcing positive / negative affinities to reach a pre-designated value as in conventional contrastive learning approaches. The authors further propose pruning positive and negative samples during backpropagation to ensure memory efficiency. Experiments demonstrate that the proposed method can outperform tested baselines in local feature matching, panoptic segmentation, and visual place recognition.

**Strengths:**

1. The paper tackles an important task in training vision foundation models, namely enforcing multi-view consistency.
2. The presentation is clear and straightforward to follow.
3. The proposed pruning scheme enables training multi-view consistent vision foundation models at a much smaller computational cost than existing approaches.

**Weaknesses:**

1. My major concern is with the experiments. First, it is unfair to compare existing vision foundation models directly against the proposed method, since the method is additionally trained using Matterport3D data. The baselines in Table 1 should also have been additionally fine-tuned on Matterport3D. Further, since the paper is proposing a generic loss function applicable to any pre-trained vision foundation model, the technical contributions would have been better elucidated if the method improved multi-view consistency for other vision foundation models as well, for example pre-trained DINOv2 features.

2. The motivation for setting the saturation threshold to 0.076 in L220 is unclear. Why is the gradient being 0.2% of the "maximum gradient of the sigmoid function" important for training models with the proposed loss?

3. The technical contributions of the paper is unclear. From a methodological point of view, the paper adapts soft average precision loss from prior literature with a threshold-based pruning to ensure memory efficiency, which is not strongly novel. Further, the experimental results are largely limited to fine-tuning a small CNN operating on top of DINO features. Therefore the scalability of the proposed training scheme is unclear.

**Questions:**

Please refer to the weaknesses section above. Below are a few additional questions:

1. What are the exact computational requirements of the paper? The only place that I could find hardware requirements was L302. Is a single RTX8000 GPU all we need?
2. What is the reason for training the model on Matterport3D? I feel datasets such as ScanNet will contain a much more diverse range of scenes. Since the only model being trained is the feature refinement CNN, training on ScanNet will not be prohibitively expensive.

**Limitations:**

Yes the limitations are stated in the supplementary material.

---

> ### Author Rebuttal · Authors · 2024-08-05
>
> Thank you for your thorough review and for highlighting both the strengths and weaknesses of our work. We appreciate your feedback, and we would like to address your concerns in detail below.
>
> ---
>
> **Weakness 1: Fairness of comparisons and generalization of the proposed method**
>
> We understand your concern regarding the comparison with existing vision foundation models. However, we would like to clarify that our training and evaluation strategy is designed to test generalization across diverse scenes and datasets, not just on Matterport3D. Specifically, we evaluate on different environments than those used in training, including environments from a different dataset (ScanNet), ensuring that our network is not simply overfitting on the training set, but instead learning transferable and generalizable features.
>
> Regarding the inclusion of a DINOv2 backbone, we have extended our experimental analysis to include results with a DINOv2 ViT-Base backbone (as described in the global author response). The results show clear performance improvements over the original DINOv2 features, which supports the general applicability of our method across different backbone architectures. This new evidence strengthens the claim that our method improves multi-view consistency in a broader context.
>
> ---
>
> **Weakness 2: Motivation for the saturation threshold $\Delta$**
>
> The threshold $\Delta$ is indeed a key hyper-parameter in our method, and its selection is crucial for balancing memory efficiency with the accuracy of the loss function approximation. The chosen value for $\Delta$ represents a trade-off: a smaller $\Delta$ leads to greater memory savings but can also introduce distortions to the loss function, affecting the training dynamics.
>
> We selected $\Delta$ such that the gradient at $\sigma(\Delta)$ is 0.2% of the maximum gradient of the sigmoid function. This choice allows us to achieve significant memory savings (by a factor of approximately 5) while minimizing the distortion of the loss gradients. In this way, we ensure that the memory efficiency gains do not come at the expense of training stability or performance.
>
> The ablations provided in Table 1 of the paper offer insight into these trade-offs, showing how different values of $\Delta$ impact both memory usage and model performance.
> We agree that the explanation for our specific choice could have been more explicit in the paper, and we appreciate your feedback on this. We believe that our current justification is grounded in practical considerations of training efficiency and accuracy, and we have provided empirical evidence to support it.
>
> ---
>
> **Weakness 3: Technical contributions and scalability concerns**
>
> We would like to emphasize that our approach is more than just an adaptation of the LoCUS architecture with threshold-based pruning. The pruning mechanism we propose is based on a detailed analysis of the gradient behavior, and it includes correction terms to ensure that the loss remains *unbiased*. We also address the bias in existing batch implementations of the general Smooth-AP loss, deriving the compensating correction terms necessary if using this loss for mini-batches, as is always the case in practice. This addresses a fundamental flaw in this loss function as used in the literature. Overall, our careful loss design addresses the challenges associated with unbiasing the loss signal, improving the memory efficiency and stabilising the gradients, making our method a significant contribution beyond a simple combination of existing techniques.
>
> Regarding scalability, we agree that larger-scale experiments would be beneficial to fully explore the potential of our approach. However, even within the computational constraints we had, we observed consistent performance improvements with increased model size, as evidenced by our experiments with the LoCUS architecture. This suggests that our method scales well, at least within the range we were able to test. We hope that future work, possibly with greater computational resources, will further validate the scalability of our approach on larger foundation models.
>
> ---
>
> **Questions and Additional Clarifications**
>
> 1. **Computational Requirements**: As noted in our paper, our experiments were conducted using a single RTX8000 GPU, with each training run taking approximately 1.5 days. This setup reflects the memory efficiency of our method, making it accessible even for researchers with limited computational resources.
>
> 2. **Choice of Matterport3D Dataset**: The method most similar to ours, LoCUS, trained its models on the Matterport3D dataset only, so we followed their choice to avoid additional confounding factors in comparing the two methods. The Matterport3D dataset is particularly suitable for our task of enforcing multi-view consistency due to its diversity and the way it captures varied viewpoints of the same scene through panorama cropping. We acknowledge that ScanNet is another valuable dataset; however, due to its trajectory-based data collection, there is less viewpoint variation per scene compared to Matterport3D, which influenced our dataset selection.
>
> ---
>
> We hope these clarifications address your concerns and provide a better understanding of the contributions and significance of our work. We appreciate your thoughtful feedback and believe that our additional explanations and results strengthen the case for our contributions. We respectfully invite you to reconsider your assessment of our work in light of the additional evidence and explanations provided.
>
> Thank you once again for your time and effort in reviewing our submission.

---

> > ### Comment · Reviewer_jhZZ · 2024-08-13
> >
> > Thank you for the response. While the rebuttal has allayed some of my concerns, I am willing to keep my initial rating. Here is why:
> >
> > 1. I acknowledge that the authors have conducted experiments regarding generalization on other datasets such as ScanNet. However, my major concern was that all the tested baselines should have been fine-tuned on the training dataset for LoCo (i.e., Matterport3D), and then be evaluated on the test set. Otherwise, the comparisons will not be fair, since only LoCo had access to the multi-view information from Matterport3D.
> >
> > 2. Regarding the motivation of the saturation threshold, I am still not clear why "0.2% of the maximum gradient of the sigmoid function" is a key desiderata for choosing its value. Where does the number "0.2%" come from? The authors' explanation is not sufficient to clarify their decision on the saturation threshold value.

---

> > > ### Author Response · Authors · 2024-08-13
> > >
> > > Thank you for considering our response and for clearly outlining your main concerns. We appreciate the opportunity to address your remaining concerns and provide further clarification below:
> > >
> > > 1. We acknowledge the concern and will include this fine-tuning experiment in the final revision; we need more time to complete this for all compared methods. However, we would like to emphasise that one of the compared methods (CroCo-v2) is already trained with a multi-view loss on a very much larger multi-view dataset (a combination of simulated Habitat images and real images from ARKitScenes, MegaDepth, 3D Street view, IndoorVL; a total of 5.3 million image pairs) than Matterport3D and so has a significant advantage over LoCo, especially on the ScanNet experiments. LoCo’s better performance on the validation tasks highlights the benefits of explicitly including location-consistency as a goal of the loss function.
> > >
> > > 2. The 0.2% threshold is arbitrary, albeit selected empirically (see Table 1 of the paper). It represents a trade-off between performance and memory-efficiency. Setting it to 0.2% of the maximum gradient leads to minimal impact on the gradient signal (and minimal impact on performance) while achieving a 5x memory reduction. In this respect it resembles other hyperparameters used in training algorithms, e.g. learning rates, where the practitioner’s choice is guided by empirical performance rather than theoretical guarantees of optimality.
> > >
> > > We hope these additional clarifications address your concerns more comprehensively. We are committed to improving our work and appreciate the constructive nature of your feedback. Thank you once again for your valuable insights.

---

### Author Rebuttal · Authors · 2024-08-06

We thank all reviewers for their careful reading and thoughtful comments. In this section, we outline the additional ablations and evaluations requested by the reviewers, which are presented in the PDF attached to this response.

## Pixel Correspondence Estimation (Table 1)
### LoCo with DINOv2 Backbone
As requested by reviewers **jhZZ** and **YTBi**, we provide results where the DINO backbone used in the original submission is replaced by the DINOv2 ViT-Base backbone ("LoCo w/ DINOv2 backbone"). The resulting features significantly outperform the original pre-trained DINOv2 features (also reported in Table 1) for finding accurate pixel correspondences, showing its advantage in tasks that require location-consistent features. However, it slightly underperforms the LoCo model trained with the DINO-ViT-Base8 backbone used in the original submission. We hypothesize that this arises from the coarser feature map of the DINOv2-ViT-Base14 backbone compared to the DINO-ViT-Base8 backbone.

### LoCo with LoCUS architecture
As requested by reviewers **YTBi** and **aW6W**, we also provide results where the network architecture used in the original submission is replaced by that used in LoCUS ("LoCo w/ LoCUS architecture"). This model outperforms the original LoCUS model for small viewpoint changes, but underperforms for image pairs with larger viewpoint changes. This illustrates that the improvements in memory efficiency do not by themselves lead to improvements in performance, but that they allow for the training of larger models and higher-dimensional feature vectors with the same computational budget, the effect of which far outweighs any performance decrease due to our loss function and training algorithm changes.

## Panoptic Scene-Stable Segmentation (Table 2)
### LoCo with DINOv2 Backbone
For this task, the LoCo model trained with the DINOv2 backbone only outperforms the original DINOv2 feature extractor on some of the metrics. This is likely attributable to the coarser feature map of this backbone, which leads to less fine-grained patch-level supervision during training.

### LoCo with LoCUS architecture
For this task, the LoCUS architecture trained with the LoCo algorithm performs worse than the original LoCUS model. For this ablation, we trained for the same number of epochs as our other LoCo models, and so cannot rule out that the vision transformer blocks in the LoCUS architecture require longer training times than the convolutional layers of the LoCo model. As before, this result illustrates the value in our efficiency improvements as they ultimately unlock the training of larger models with larger feature dimensions.

---

### Decision · Program_Chairs · 2024-09-25

**Decision:**

Accept (poster)

**Comment:**

This paper received mixed reviews. While reviewers praise the interesting ideas, efficiency and good presentation, there were concerns around potentially unfair comparisons and unjustified hyperparameters. The rebuttal clarified several concerns, and there was a good discussion between the reviewers to weigh the pros and cons. Even though there was no final consensus on the ratings, the AC recognizes that the work can be accepted into NeurIPS, given all its positive aspects and the additional information provided in the rebuttal. We require that the clarifications and all relevant information provided in the rebuttal be incorporated in the camera-ready version.